# Pyrolytic Behavior of Major Biomass Components in Waste Biomass

**DOI:** 10.3390/polym11020324

**Published:** 2019-02-13

**Authors:** Haoxi Ben, Zhihong Wu, Guangting Han, Wei Jiang, Arthur Ragauskas

**Affiliations:** 1Southeast University, Nanjing 210096, China; 18915408402@163.com; 2Key Laboratory of Energy Thermal Conversion and Control of Ministry of Education, Nanjing 210096, China; 3Qingdao University, Qingdao 266071, China; hanguangting@gmail.com (G.H.); jiangwei@gmail.com (W.J.); 4State Key Laboratory of Bio-Fibers and Eco-Textiles, Qingdao University, Qingdao 266071, China; 5Joint Institute for Biological Sciences, Biosciences Division, Oak Ridge National Lab, Oak Ridge, TN 37831, USA; aragausk@utk.edu; 6Department of Chemical and Biomolecular Engineering, University of Tennessee, Knoxville, TN 37996, USA

**Keywords:** tannin, hemicellulose, waste biomass, HSQC-NMR, pyrolysis mechanism

## Abstract

The pyrolytic behavior of several biomass components including cellulose, hemicellulose, lignin, and tannin, from two sources of waste biomass (i.e., pine bark and pine residues) were examined. Compared to the two aromatic-based components in the biomass, carbohydrates produced much less char but more gas. Surprisingly, tannin produced a significant amount of water-soluble products; further analysis indicated that tannin could produce a large amount of catechols. The first reported NMR chemical shift databases for tannin and hemicellulose pyrolysis oils were created to facilitate the HSQC analysis. Various C–H functional groups (>30 different C–H bonds) in the pyrolysis oils could be analyzed by employing HSQC-NMR. The results indicated that most of the aromatic C–H and aliphatic C–H bonds in the pyrolysis oils produced from pine bark and pine residues resulted from the lignin and tannin components. A preliminary study for a quantitative application of HSQC-NMR on the characterization of pyrolysis oil was also done in this study. Nevertheless, the concepts established in this work open up new methods to fully characterize the whole portion of pyrolysis oils produced from various biomass components, which can provide valuable information on the thermochemical mechanisms.

## 1. Introduction

Growing concerns about increasing global energy consumption [1] and the effects of growing carbon dioxide emissions from fossil fuels has reinvigorated interest in sustainable energy and chemical sources. Biomass is a promising renewable resource for the sustainable production of fuels and chemicals that, to date, are made primarily from fossil resources [2,3]. Lignocellulosic biomass contains three major constituents: cellulose, hemicelluloses, and lignin. Several reviews summarized the distribution of these major biopolymers in several hardwoods, softwoods, and agricultural residue species [4,5]. However, in softwood select tissues, including leaves, needles, and barks, there is another type of biopolymer—tannins—which could be even more abundant than lignin occasionally [6]. There are two types of tannins; hydrolyzable tannins are derivatives of gallic acid, which are esterified to polyols such as glucose (left structure in Figure 1), and condensed tannins (right structure in Figure 1), which are polymers of flavonoids, and are much more complicated than hydrolyzable tannins. Hernes et al. [6] conducted very detailed research on tannin signatures of 117 tissues from 77 different biomass species at the molecular level. Surprisingly, compared to the pyrolysis of cellulose and lignin, there are only very limited references which investigated the pyrolysis of tannins. Ohara et al. [7] examined the pyrolysis of several tannin model compounds, including catechin and epicatechin. The authors indicated that catechol and 4-methylcatechol are the major products of pyrolysis of catechin and epicatechin model compounds, and they proposed possible formation pathways for these two products. Gaugler et al. [8] used thermogravimetric analysis (TGA) to analyze the thermal degradation of various condensed tannins and tannin model compounds. The highest weight loss of catechin occurred at 197 °C, while that of sulfited tannin occurred at 159 °C, that of tannin acetate occurred at 189 °C, and that of Quebracho tannin (commercial tannin extracted from Quebracho) occurred at 271 °C. Reported [7] pyrolysis products from tannins are summarized in Appendix A.

The current annual stock of waste biomass is estimated as ~46 exajoules (EJ) from agricultural biomass and ~37 EJ from forestry biomass on a worldwide basis, totaling approximately 83 EJ, which represents approximately 20% of total worldwide energy consumption. Economic and practical limitations, including collection and transportation, are still holding up the price for the applications of these waste bioresources. Based on the United States (US) billion-ton update, the currently (2012) available forestry waste at <$60 per dry ton is estimated to be ~90 million dry tons in the US. In the meanwhile, the currently available agricultural residues and other waste resources at the same cost are suggested to be ~240 million dry tons in the US [9]. Thermal deconstruction is a promising approach to convert these sources of waste biomass to chemicals and precursors of biofuel [10]. Loblolly pine (*Pinus taeda*) is one of the abundant softwood species in the southeastern US and is widely used in various industries [5]. Harvesting operations leave a large amount of residues (e.g., stumps, limbs, tops, and dead trees, which are usually referred to as slash) and bark, which represent an important source of energy and chemicals [11]. Various researchers examined the thermal decomposition process for barks and residues [11,12,13,14,15,16].

For example, Pakdel et al. [11] pyrolyzed a mixture of spruce, fir, and pine, and maple barks and residues at different temperatures (450–525 °C). They found that the highest yield of pyrolysis oil was 73.6% for softwood residue and 57.1% for softwood bark at 525 °C. The yield of pyrolysis products for the hardwood was found to be similar to that of softwood. Ingram et al. [12] fast pyrolyzed pine wood and bark, and oak wood and bark in a continuous auger reactor at 450 °C. They used GC–MS to analyze pyrolysis products, which indicated that furfural, phenol, 3-methylphenol, catechol, 3-methylcatechol, and levoglucosan were the major products in the pine bark pyrolysis oil. Arpiainen et al. [13] used a fluidized-bed reactor to pyrolyze pine bark from 400–800 °C. They found that the highest yields of pyrolysis oil (~50%) were obtained at 500 °C. The secondary reactions were reported to occur at higher treatment temperature, which lead to a decreasing yield of pyrolysis oil. Lomax et al. [14,15] examined the pyrolysis of two different pine barks (i.e., *Pinus contorta* and *Pinus radiata*) from 505–840 °C. Using GC–MS analysis, the qualitative characterization for pyrolysis oils indicated that furfural, catechol, guaiacol, 2-ethylphenol, and 2-hydroxy-5-methyl phenol were the major components [16].

Due to the limitation of volatility of high-molecular-weight components in the pyrolysis oil, it was reported that only about 40% of pyrolysis oil could be detected by GC [10]. Therefore, many researchers have tried finding alternative characterization methods, which could analyze the whole portion of pyrolysis oil. Characterizations of pyrolysis products using high-resolution mass spectrometry, including both quadrupole time-of-flight mass spectrometry (Q-TOF MS) and Fourier-transform ion cyclotron resonance mass spectrometry (FT-ICR MS), were proposed by several researchers [17,18,19,20,21]. Several different pyrolysis oils produced from different whole sources of biomass, such as eucalyptus, eucalyptus bark, cellulosic mud, water hyacinth, pine wood, birch wood, algae, and switchgrass, using different thermochemical processes including fast pyrolysis, slow pyrolysis, and hydrothermal liquefaction, were analyzed by employing high-resolution mass spectrometry. FT-ICR MS was reported to be a promising method to analyze pyrolysis oil due to its unsurpassed resolution and accuracy [17]. However, the extremely complicated components in the pyrolysis oil (~2000–5000 peaks can be detected using FT-ICR MS) bring many barriers for this method [20]. The results from characterization of pyrolysis oil using FT-ICR MS can be presented as C_x_, N_x_, and O_x_ species. However, a detailed investigation of different chemical functional groups in the pyrolysis oils calls for other analytical methods. In a recent study, quantitative ^13^C-NMR combined with comprehensive two-dimensional gas chromatography (GC × GC) was used to characterize fast pyrolysis bio-oils, which provided new information on the chemical composition of bio-oils for further upgrading [22]. The use of different NMR techniques to characterize pyrolysis oils can accomplish this task, and is also a rapidly growing field of study [23,24,25,26,27,28,29]. The usage of HSQC-NMR (heteronuclear multiple quantum correlation nuclear magnetic resonance) provides a complete analysis of bio-oils, and the chemical shift assignments of different types of C–H bond presented in the pyrolysis oils produced from lignin and cellulose were readily identified. In this study, the detailed characterization of pyrolysis oils obtained from tannin and hemicellulose were accomplished using HSQC-NMR. The fingerprint analysis of these biomass components will open up a new way of understanding and utilizing pyrolysis products from biomass.

## 2. Experimental Section

### 2.1. Materials and Methods

All reagents used in this study were purchased from VWR International or Sigma-Aldrich (St. Louis, MO, USA) and used as received. Loblolly pine (*Pinus taeda*) bark and residue (mixture of stumps, limbs, tops, and dead trees) were collected from a University of Georgia research plot in Macon, GA [5]. The wood samples (Appendix A) were refined with a Wiley mill through a 0.13-cm screen and dried under high vacuum at 50 °C for 48 h, and stored at ~0 °C prior to use. Cellulose was commercially purchased from Sigma-Aldrich. Xylan was also commercially purchased from Sigma-Aldrich, which is produced from oak wood, and was used to present hemicellulose in this study.

### 2.2. Tannin Extraction [30]

The tannin used in this study was obtained from *Pinus radiata* bark, which was collected in the MADECAM saw mill, located in the Concepción Province, Bio Region, Chile. The bark was obtained with a log debarker from 15-year-old and 20-year-old trees. The time since harvesting was less than one month and the debarking occurred during the last 24 h. The bark was treated in a slow-speed shredder, with a 20-mm cutting gap, and was dried under atmospheric conditions up to a water content of 20–30%.

The extraction of pine bark tannin was done in two sequential extractions steps, with a 75% methanolic solution (*w*/*w*) at 120 °C for 2 h. The details of this procedure involved a 4-m^3^ conical stainless-steel reactor being filled with the equivalent of 300 kg of dry bark. Concentrated methanol and water were added, to achieve a methanol concentration of 75% (*w*/*w*) and a solid/liquid relationship of 1/5 (*w*/*w*). The solution was continuously pumped through an external head exchanger, fed with 500 kPa. The temperature of the solution was heated up from ambient conditions to 120 °C in 4 h. This maximum temperature was maintained for 120 min. Thereafter, the solution was cooled, by opening a relief valve, to 80 °C in 40 min. The day after the solution was drained from the reactor, fresh methanol and water were added to achieve the same concentration and solid/liquid ratio as in the first extraction step. The same procedure was repeated in an analogous manner.

The methanolic solutions with the dissolved bark from the first and second extraction steps were combined and evaporated from 4–5% to 15–20% solid content. During the evaporation, the methanol was preferentially removed from the solution yielding an aqueous slurry. The insoluble material was separated by decantation, and the water-soluble phase was concentrated further and spray-dried. The water-soluble tannin was dried under high vacuum at 50 °C for 48 h and stored at ~0 °C prior to use. The tannin used in this study was characterized by quantitative ^13^C NMR (Appendix A).

### 2.3. Kraft Pulping

The softwood kraft pulping liquor used was prepared using a conventional method [31]. Loblolly pine was employed as the wood source for the pulping process. Some relevant preparation conditions are presented in Table 1.

### 2.4. Lignin Separation and Purification [23]

Lignin was isolated from a softwood (pine wood) kraft pulping liquor using the following methods: in brief, the cooking liquor was filtered through filter paper and the filtrate was treated with ethyenediaminetetraacetic acid (EDTA)–2Na^+^ (0.50 g/100.00 mL liquor) and stirred for 1 h. The liquor was adjusted to a pH value of 6.0 with 2.00 M H_2_SO_4_ and stirred vigorously for 1 h. The liquors were then further acidified to a pH of 3.0 and frozen at −20 °C. After thawing, the precipitates were collected on a medium sintered glass funnel and washed three times with cold water by suspending the precipitates in the water and stirring vigorously at 0 °C for 1 h. The precipitates were collected, air-dried, and Soxhlet-extracted with pentane for 24 h. The solid product was air-dried and further dried under high vacuum at 45 °C for 48 h. The resulting purified kraft lignin sample was stored at −5 °C. The lignin used in this study was characterized by quantitative ^13^C NMR (Appendix A).

### 2.5. Equipment and Process of Pyrolysis [23]

Pyrolysis experiments were conducted in a quartz pyrolysis tube heated with a split-tube furnace. Typically, the pyrolysis sample (4.00 g) was placed in a quartz sample boat that was then positioned in the center of a pyrolysis tube. A K-type thermal couple was immersed in the sample powder during the pyrolysis to measure the heating rate (the heating rate was ~2.7 °C/s). The pyrolysis tube was flushed with nitrogen gas and the flow rate was adjusted to a value of 500 mL/min, and then the tube was inserted into the pre-heated furnace. The outflow from pyrolysis was passed through two condensers, which were immersed in liquid N_2_. Upon completion of pyrolysis, the reaction tube was removed from the furnace and allowed to cool to room temperature under constant N_2_ flow. The condensers were then removed from liquid nitrogen. The pyrolysis char and oil were collected for subsequent chemical analysis. In general, the liquid products contained two immiscible phases, referred to as heavy and light oil. The light oil was acquired by decantation, and the heavy oil was recovered by washing the reactor with acetone, followed by evaporation under reduced pressure. Char yields were determined gravimetrically, and gas formation was calculated by mass difference.

### 2.6. Characterization of Pyrolysis Oil by HSQC-NMR [24]

All NMR spectral data reported in this study were recorded with a Bruker Avance/DMX 400 MHz NMR spectrometer. HSQC-NMR spectra were acquired using 70.0 mg of pyrolysis oil (combined light oil and heavy oil by dissolving of two immiscible phases in acetone, followed by evaporation under reduced pressure) dissolved in 450 µL of dimethyl sulfoxide (DMSO-*d*_6_), employing a standard Bruker pulse sequence “hsqcetgpsi.2” with a 90° pulse, 0.11-s acquisition time, a 1.5-s pulse delay, a ^1^J_C–H_ of 145 Hz, 48 scans, and the acquisition of 1024 data points (for ^1^H) and 256 increments (for ^13^C). The ^1^H and ^13^C pulse widths were p1 = 11.30 µs and p3 = 10.00 µs, respectively. The ^1^H and ^13^C spectral widths were 13.02 ppm and 220.00 ppm, respectively. The central solvent peak was used for chemical shift calibration. HSQC-NMR data processing and plots were carried out using MestReNova v7.1.0 software’s default processing template and automatic phase and baseline correction.

## 3. Results and Discussion

To further understand the pyrolytic behavior of whole biomass, the thermal decomposition process for several biomass components including cellulose, hemicellulose, lignin, and tannin was examined at 600 °C, which was found as the optimal temperature to yield the highest amount of pyrolysis oils (Table 2 and Table 3). The yields of the light oil, heavy oil, char, and gas for the pyrolysis of these major components are summarized in Table 1. The results show that, compared to the two aromatic-based components (lignin and tannin), carbohydrates including cellulose and hemicellulose produced much less char but more gas, which indicates that carbohydrates are relatively readier to decompose than lignin and tannin. In addition, lignin produced mostly organic products (heavy oil) compared to the other biomass components, which represents a potential resource for a biofuel precursor. Surprisingly, tannin produced a significant amount of water-soluble products (light oil). Further analysis of the composition of tannin pyrolysis oil indicated that tannin could produce a large amount of catechols, which are water-soluble and could be used as a bio-chemical resource. Hemicellulose was reported [24] to be the very first biomass component to decompose during thermal treatment, which supports the results shown in Table 1 that hemicellulose produced most gas products compare to the other biomass components.

The pyrolysis process for the two sources of waste biomass (pine bark and pine residues) were examined at different temperatures, and the results are shown in Table 3 and Table 4. The results indicate that the yield of char decreased, and the yield of gas increased at higher reactor temperatures for both biomasses. Compare to the pine bark, pine residue produced more (up to 15 wt.% of dry biomass) pyrolysis oils under the same conditions. The higher yields of pyrolysis oils are due to the higher carbohydrate (cellulose and hemicellulose) contents in pine residue (65.9%) than the pine bark (50.9%) [5], which could produce more pyrolysis oils than lignin and tannin.

Due to the limitation of GC–MS for the characterization of pyrolysis products, many researchers have pursued alternative characterization methods, which could analyze the whole portion of pyrolysis oil, such as NMR. HSQC-NMR exhibits a superior ability to deal with such complicated mixtures than other NMR methods. The strengths of this methodology include less spectral overlap problems and reduced NMR acquisition time. In this study, the first reports of NMR chemical shift databases for tannin and hemicellulose pyrolysis oils were created to facilitate this analysis. The detailed information of the databases can be found in the Appendix A as Appendix A. Similar NMR databases for lignin and cellulose pyrolysis products were reported previously in the literature [24].

The aromatic C–H bonds, levoglucosan, methoxyl groups, and aliphatic C–H bonds in the HSQC-NMR spectra for the pyrolysis oils produced from cellulose, hemicellulose, lignin, and tannin at 600 °C are shown in Figure 2a–d. Figure 2a shows that there were substantially more aromatic products in lignin and tannin pyrolysis oils than in cellulose and hemicellulose pyrolysis oils, which is also consistent with the yield results (more heavy organic products in lignin and tannin pyrolysis oils) shown in Table 1. In addition, Figure 2a also indicates that hemicellulose produced relatively more aromatic products than cellulose, which is also consistent with yield results and a literature report [32]. Compared to lignin pyrolysis oil, tannin pyrolysis oil contains a large amount of catechols and a relatively lower content of mono-hydroxyl phenols, which is consistent with the GC–MS analysis and ^13^C NMR results (Appendix A; Figure 3). Ohara et al. [7] also reported a similar result, where they used pyrolysis gas chromatography to analyze the pyrolysis products from the model compounds of tannin. Catechol and 4-methylcatechol were detected as the major components in the pyrolysis oil produced from procyanidin (PC, structure shown in Figure 3), which is the major type of tannin present in pine barks. The formation of catechol was attributed to the cleavage of C1’ and C2 in the PC structure, and 4-methylcatechol was formed via the fission of the pyran ring, followed by the cleavage of the ether bond. The possible pathways of the formation of these two major components in the tannin pyrolysis oil are shown in Figure 3. Figure 2d shows that the major type of aliphatic C–H bonds belonged to methyl-aromatic carbon, which is also evidence that 4-methylcatechol was one of the major components in the tannin pyrolysis oil. The large amount of light oil produced from tannin indicated that there was a significant amount of water generated during the pyrolysis process, which could be attributed to the dehydration of hydroxyl groups in the pyran ring. As expected, levoglucosan (Figure 2b) was the major component in the cellulose pyrolysis oil. Lignin and tannin did not produce any similar structures during the thermal decomposition process. A very similar dehydrated five-carbon monosaccharide was found in the hemicellulose pyrolysis oil, which indicates that similar decomposition pathways to those seen with cellulose also occur during the decomposition of hemicellulose. Figure 2c indicates that methoxyl groups only appeared in the lignin pyrolysis oil. Two different types of methoxyl groups indicate the rearrangement process during the thermal treatment of lignin [24]. Figure 2d shows that, compared to the other biomass components, hemicellulose produced a significant amount of aliphatic C–H bonds, which could be assigned as methyl groups in the furan ring (could be produced from deep dehydration/decomposition of the five-carbon monosaccharide), α and β positions of a carbonyl group (could be produced from uncompleted dehydration/decomposition of the five-carbon monosaccharide), and an aliphatic chain (could be produced from the cleavage of side chains of hemicellulose). Similar results were reported by Patwardhan et al. [32]. In their study the pyrolytic behaviors of switchgrass hemicellulose were examined, and 2-methyl furan and several anhydro xyloses were found as major components in the pyrolysis oil. On the basis of Figure 2d, lignin and tannin pyrolysis oils had a very similar group of aliphatic C–H bonds, which represent the methyl groups in the aromatic rings. Compared to the other biomass components, cellulose produced a very limited amount of aliphatic C–H bonds, and most of them could be assigned as α and β positions of a carbonyl group.

A detailed understanding of the chemical structures in the pyrolysis oils produced from major biomass components is anticipated to facilitate the characterization of whole-biomass pyrolysis oils. Figure 4a,b show the HSQC-NMR spectral data and the assignments of each carbon in the levoglucosan, which was present in the pine bark and residue pyrolysis oils [33], as the intensity of those peaks indicated that levoglucosan was one of the major products. The content of levoglucosan slightly increased with elevated pyrolysis temperatures, which is consistent with a literature report [24]. Similarly, most of other peaks in this area also increased with a higher reactor temperature, which could represent similar dehydrated five-carbon monosaccharide produced from hemicellulose. Therefore, a higher thermal decomposition temperature will favor the production of a more dehydrated monosaccharides. Figure 4c,d show the aromatic C–H bonds in the HSQC-NMR spectra for the pine bark and residues pyrolysis oils. The results indicate that the major aromatic components in the pine bark and residue pyrolysis oils contained catechol, guaiacol, and phenol types of aromatic C–H bonds, which was comparable with the lignin and tannin pyrolysis oils. There was a relatively more furfural type of aromatic C–H bonds in the pine residue pyrolysis oil than in the pyrolysis oil produced from pine bark, which indicated that there were more carbohydrate decomposition products in the pine residue pyrolysis oil. Huang et al. [5] indicated that there were more carbohydrates in pine residue (65.9%) than pine bark (50.9%), but less lignin (pine residue 26.7%, pine bark 33.7%) and tannin (pine residue 3.7%, pine bark 11.6%), which supports the results shown in Figure 4c,d. Very interestingly, the influences from different temperatures on the aromatic products appear to be very limited, which may be due to the relatively thermal stable structures for these aromatic products. The slight increase in the number of aromatic C–H bonds in the phenolic structure at a higher temperature may have been due to the decomposition of side chains. Figure 4e,f indicate that there were more rearranged methoxyl groups with no hydroxyl or ether bond in the *ortho* position of the methoxyl group than the native type (a hydroxyl or ether bond in the *ortho* position of the methoxyl group) in the pyrolysis oils produced from pine bark and residue, indicating that such a rearrangement of methoxyl groups also occurred during the pyrolysis of whole biomass. Figure 4g,h show that the contents of aliphatic C–H bonds slightly increased with the elevated reactor temperatures, and most of those aliphatic C–H bonds (α and β positions in aromatic ring) resulted from the pyrolysis of lignin and tannin components in the pine bark and pine residue, indicating that these aromatic biomass components require a relatively higher temperature to further decompose into small fragments. Figure 3 shows the formation pathways for one of the major pyrolysis products from tannin, 4-methylcatechol. The pyrolysis of lignin was also reported [23,24] to produce a significant amount of methyl-aromatic bonds via a rearrangement of methoxyl groups. Due to the relatively high lignin and tannin contents in the pine bark and residue, the aliphatic C–H bonds in the α and β positions in aromatic rings could be expected as one of the major types of aliphatic bonds in the pyrolysis oils produced from pine bark and residue. To further investigate the pyrolytic behaviors of biomass components, several comparisons between artificial spectra (a mixture of pyrolysis oils produced from lignin, tannin, cellulose, and hemicellulose) and whole biomasses were examined and the results are shown in Appendix A. The results show that the artificial spectra could represent whole-biomass pyrolysis oil very well, whereby almost all the major peaks in the real pyrolysis oils could be found in the artificial spectra, indicating that the technique developed in this study could be employed as a new method to fully characterize the whole portion of pyrolysis oils produced from various biomass components and whole sources of biomass, and to facilitate the high-throughput screening of pyrolysis oils. Very interestingly, compared to the artificial spectra, only a very limited number of “new” peaks could be found in the aliphatic and dehydrated monosaccharide ranges of whole-biomass pyrolysis oils, indicating that there may only have been very limited cross-link reactions between biomass components. The products from these possible cross-link reactions could have become the “new” side chains in the major pyrolysis components.

Several experimental approaches [34,35,36] for quantitative HSQC-NMR characterization were reported; however, they all involved special pulse sequences and prolonged experimental time. In this study, a preliminary investigation for a quantitative application of normal HSQC-NMR on the characterization of pyrolysis oil, produced from a pilot plant reactor [37], was also accomplished. Several model compounds for the pyrolysis oil, including guaiacol, levoglucosan, toluene, anisole, and *n*-propylbenzene were used to create calibration curves. The HSQC-NMR results for the standard mixtures and the pyrolysis oil produced from a pilot plant reactor are shown in Appendix A. Aromatic C–H, levoglucosan C–H, methoxyl C–H, and aliphatic C–H bonds in the standard mixtures were analyzed and the integrations for these C–H bonds in different standards with different concentrations were plotted as shown in Appendix A to create calibration curves. To evaluate the calibration curves, three artificial mixtures of pyrolysis oil with model compounds (guaiacol, levoglucosan, and *n*-propylbenzene) were examined by HSQC-NMR. The calculated concentrations for different C–H bonds agreed with the designed values in these artificial mixtures (Appendix A). To further evaluate the proposed method, quantitative ^13^C NMR, and DEPT 135 and DEPT 90 were also employed to further characterize the pyrolysis oil (Appendix A). The use of chemical shift assignment ranges (Appendix A) for pyrolysis oils in the ^13^C NMR were previously reported in the literature [23,38]. The quantitative ^13^C NMR used in this study was examined using two commercial petroleum standard mixtures (Appendix A, which can represent complicated mixtures, such as petroleum oils or pyrolysis oils. The results shown in Appendix A indicate that the employed quantitative ^13^C NMR can provide precise analysis for different functional groups and different carbons for very complicated mixtures. To ensure the accuracy, the T1 values for different functional groups in the pyrolysis oil were measured (Appendix A). The further characterization of pyrolysis oil (produced from a pilot plant reactor) and artificial mixtures were analyzed using quantitative ^13^C NMR (Appendix A). The results indicate that the quantitative ^13^C NMR procedure used in this study can provide precise data for different functional groups in the pyrolysis oil using the internal standards. Furthermore, the quantitative data for different aromatic C–H, levoglucosan C–H, methoxyl C–H, and aliphatic C–H bonds in the pyrolysis oil proposed using calibration curves and HSQC-NMR are shown in Appendix A, and comparable results with quantitative ^13^C NMR were found, indicating that the proposed method can provide quantitative data on different C–H bonds in the pyrolysis oils. Another interesting result is that, based on the HSQC NMR data, there were 6.46 µmol of aliphatic C–H bonds in 1 mg of pyrolysis oil, while the quantitative ^13^C NMR indicated that there were 2.58 µmol of aliphatic carbons (in aliphatic C–C bonds) in 1 mg of pyrolysis oil, indicating that each aliphatic carbon in this pyrolysis oil should attach an average of 6.46/2.58 = 2.5 protons. Due to the limited overlaps between aliphatic C–C and C–O bonds in ^13^C NMR, it is highly possible that a similar result can also be provided by employing quantitative ^13^C NMR, and DEPT 135 and DEPT 90. The proposed CH_x_ number using such a method is 2.6, which is very close to the result (x = 2.5) correlating both ^13^C NMR and HSQC-NMR investigations, which is also further evidence that the proposed HSQC-NMR method can provide a precise analysis of different C–H bonds in the pyrolysis oil.

## 4. Industrial Application of Pyrolysis Bio-Oils

Pyrolysis bio-oils are mixtures of about 200 organic compounds. It was reported that all determined compounds can be classified into nine groups: furans, aldehydes, ketones, phenols, acids, benzenes, alcohols, alkanes, and polycyclic aromatic hydrocarbons (PAHs) [39]. The characterized chemical compounds of bio-oils are used in different industrial fields due to their unique properties [40]. Catechol obtained from the pyrolysis of tannin has a very high added value and is widely used in the fields of rubber, electroplating, antiseptic, sterilization, etc. Moreover, furfural, a natural dehydrating product of five-carbon sugars (e.g., arabinose and xylose) from hemicellulose biomass [41], regained attention as a bio-based alternative for the production of everything from antacids and fertilizers to plastics and paints [42]. It was reported that levoglucosan can be used as a specific molecular indicator to trace biomass combustion in sediments [43]. The application of this novel molecular tracer may bring benefits to research areas such as paleoecology, archeology, and environmental science [43]. Furthermore, aromatic hydrocarbons have good rubber compatibility, high temperature resistance, and low volatility, and they significantly improve the processing properties of rubber. Therefore, they attracted wide attention in the field of reclaimed rubber and various rubber products.

## 5. Conclusions

In this study, HSQC-NMR was utilized to characterize the whole portion of pyrolysis oils produced from various biomass components including cellulose, hemicellulose, lignin, and tannin. The fingerprint analysis of HSQC-NMR spectral data could provide semi-quantitative chemical shift assignments of more than 30 different types of C–H bond present in the pyrolysis oils. Catechols were found as major components in the tannin pyrolysis oils. Levoglucosan was the major component in the cellulose pyrolysis oil. A very similar dehydrated five-carbon monosaccharide was found in the hemicellulose pyrolysis oil, indicating that hemicellulose has a similar decomposition pathway to that of cellulose. Compared to the other biomass components, hemicellulose produced a significant amount of aliphatic C–H bonds, which could be assigned as methyl groups in the furan ring, α and β positions of a carbonyl group, and aliphatic chains. The HSQC-NMR analysis results also indicated that most of the aromatic C–H and aliphatic C–H bonds in the pyrolysis oils produced from pine bark and pine residue resulted from the lignin and tannin components. A preliminary study of the quantitative application of HSQC-NMR on the characterization of pyrolysis oil was also accomplished in this study, and the results indicate that, by employing the proposed HSQC-NMR method, a precise analysis of different C–H bonds in the pyrolysis oil can be provided. Nevertheless, the concepts established in this work open up new methods to fully characterize the whole portion of pyrolysis oils produced from various biomass components and whole sources of biomass, which could facilitate the high-throughput screening of pyrolysis oils.

## Figures and Tables

**Figure 1 polymers-11-00324-f001:**
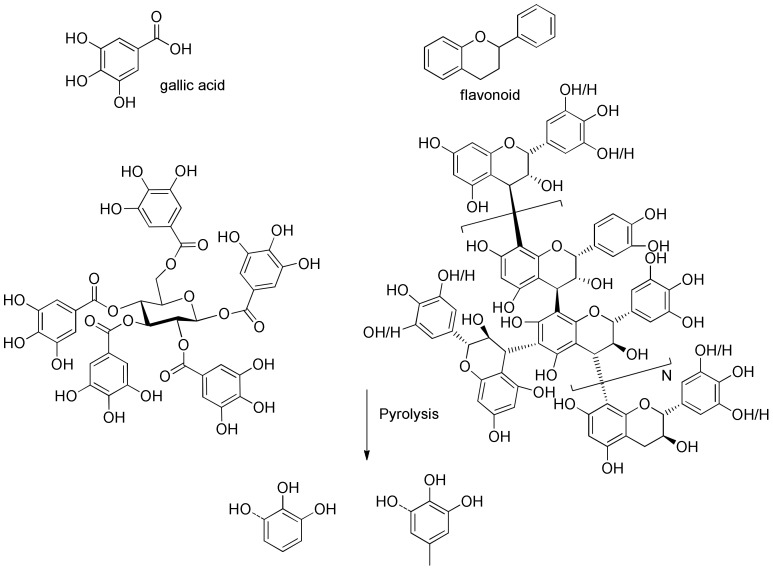
Model structures of tannin, and major pyrolysis products from tannin.

**Figure 2 polymers-11-00324-f002:**
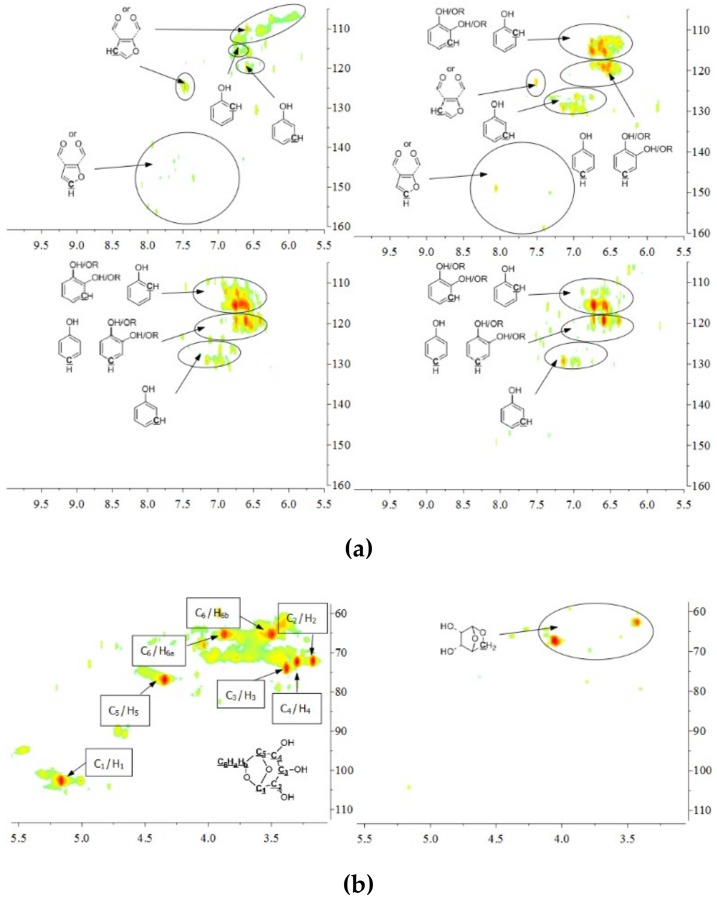
(**a**) Aromatic C–H bonds in HSQC-NMR spectra for the pyrolysis oils produced from cellulose, hemicellulose, lignin, and tannin (from top left to bottom right) at 600 °C. (**b**) Levoglucosan and a similar dehydrated monosaccharide in HSQC-NMR spectra for the pyrolysis oils produced from cellulose and hemicellulose (from left to right) at 600 °C. As expected, there would be no similar components in lignin and tannin pyrolysis oils, if samples were purified. (**c**) Methoxyl groups in HSQC-NMR spectra for the pyrolysis oils produced from lignin at 600 °C. As expected, methoxyl groups were only present in lignin pyrolysis oil, if samples were purified. (**d**) Aliphatic C–H bonds in HSQC-NMR spectra for the pyrolysis oils produced from cellulose, hemicellulose, lignin, and tannin (from top left to bottom right) at 600 °C.

**Figure 3 polymers-11-00324-f003:**
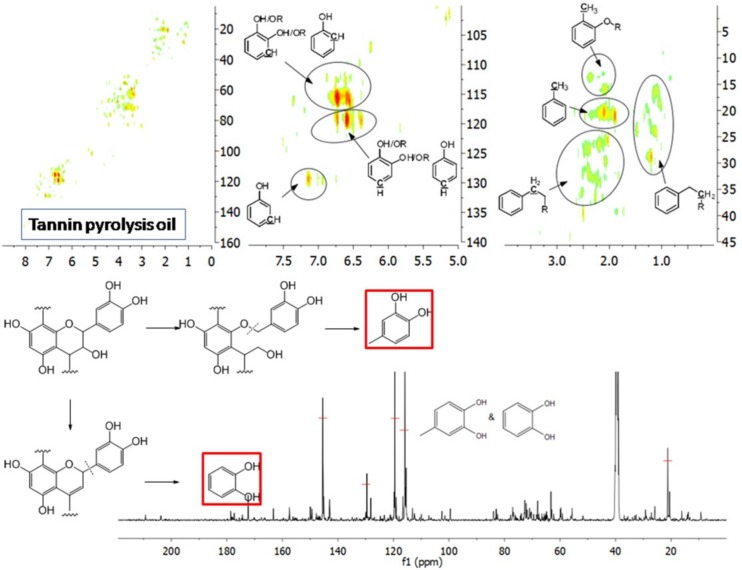
Major pyrolysis products produced from tannin and possible pathways.

**Figure 4 polymers-11-00324-f004:**
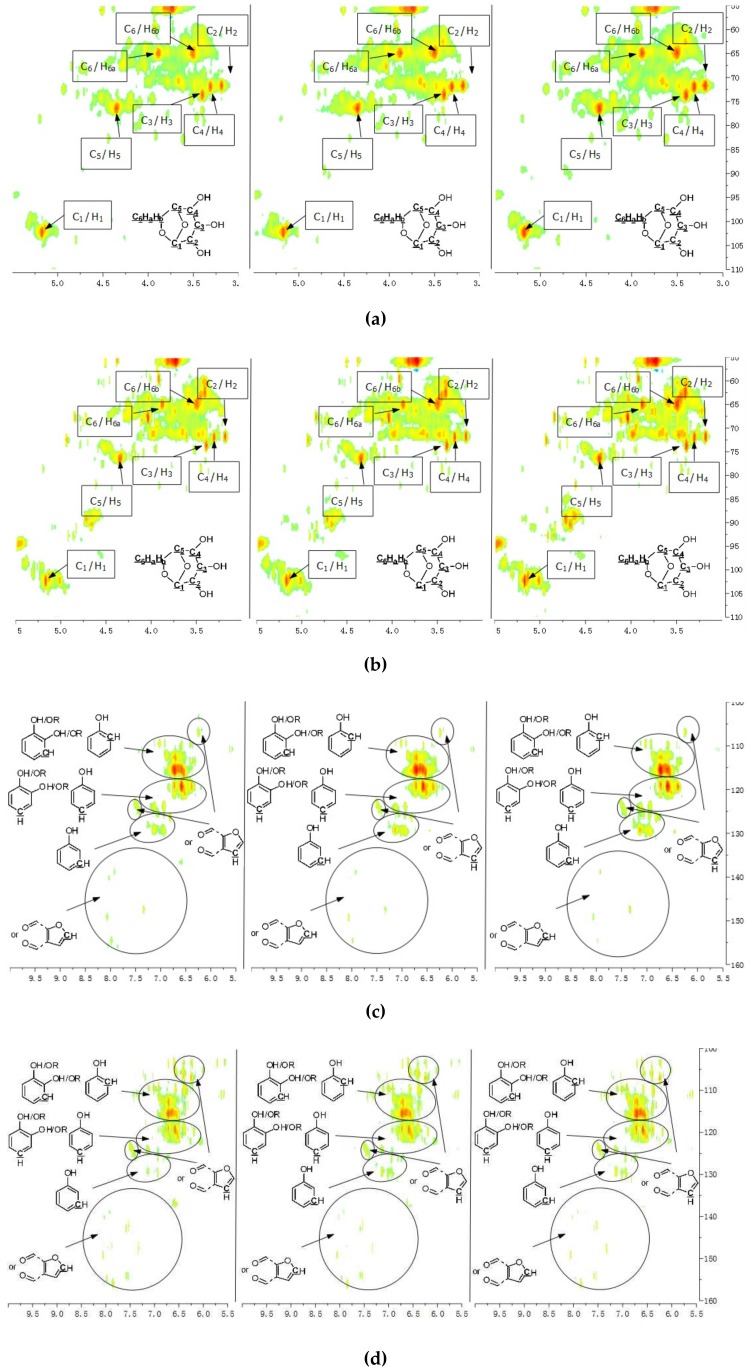
HSQC-NMR spectra and the assignments of each carbon in the levoglucosan present in the pyrolysis oils (**a**) produced from the pyrolysis of pine bark from 400 to 600 °C (from left to right), (**b**) produced from the pyrolysis of pine residue from 400 to 600 °C (from left to right). (**c**) Aromatic C–H bonds in the HSQC-NMR spectra for the pyrolysis oils produced from the pyrolysis of pine bark from 400 to 600 °C (from left to right). (**d**) Aromatic C–H bonds for the pyrolysis oils produced from the pyrolysis of pine residue from 400 to 600 °C (from left to right). (**e**) Methoxyl groups in the HSQC-NMR spectra for the pyrolysis oils produced from the pyrolysis of pine bark from 400 to 600 °C (from left to right). (**f**) Methoxyl groups for the pyrolysis oils produced from the pyrolysis of pine residue from 400 to 600 °C (from left to right). (**g**) Aliphatic C–H bonds in the HSQC-NMR spectra for the pyrolysis oils produced from the pyrolysis of pine bark from 400 to 600 °C (from left to right). (**h**) Aliphatic C–H bonds for the pyrolysis oils produced from the pyrolysis of pine residue from 400 to 600 °C (from left to right).

**Table 1 polymers-11-00324-t001:** Conventional pulping conditions [31].

Kraft Pulping	Conventional
Sulfidity, %	34.6
Effective Alkali, %	19.7
Impregnation	19.7
Temperature, °C	170
Time, min	95

**Table 2 polymers-11-00324-t002:** Yields of light oil, heavy oil, char, and gas for the pyrolysis of cellulose, hemicellulose, lignin, and tannin at 600 °C.

Biomass Components	Light Oil	Heavy Oil	Total Pyrolysis Oil	Char	Gas
Cellulose ^[a]^	58.83	10.47	69.30	11.17	19.53
Hemicellulose	36.13	13.49	49.26	23.03	27.35
Lignin ^[a]^	14.20	30.01	44.21	40.48	15.31
Tannin	37.85	9.11	46.96	40.33	12.71

^[a]^ Based on a literature report [24]. The analytical methods (GC–MS, elemental analysis, and gel permeation chromatography (GPC)) used to characterize these pyrolysis oils can also be found in the literature [23,24].

**Table 3 polymers-11-00324-t003:** Yields of light oil, heavy oil, char, and gas for the pyrolysis of bark at 400, 500, and 600 °C.

Pyrolysis Temperature (°C)	Light Oil	Heavy Oil	Total Pyrolysis Oil	Char ^[a]^	Gas
400	12.95	27.65	40.60	48.72	10.68
500	15.67	30.84	46.51	39.53	13.96
600 ^[b]^	20.23	30.65	50.88	34.58	14.54

^[a]^ The ash percentage (*w*/*w*) of bark is 0.9%. ^[b]^ Based on a literature report [4], a higher temperature above 600 °C will produce more gas products but fewer liquid products; therefore, the highest temperature used in this study was 600 °C.

**Table 4 polymers-11-00324-t004:** Yields of light oil, heavy oil, char, and gas for the pyrolysis of pine residue at 400, 500, and 600 °C.

Pyrolysis Temperature (°C)	Light Oil	Heavy Oil	Total Pyrolysis Oil	Char ^[a]^	Gas
400	25.01	30.43	55.44	33.26	11.30
500	26.45	31.95	58.40	26.02	15.58
600	26.16	34.88	61.04	22.29	16.67

^[a]^ The ash percentage (*w*/*w*) of pine residue is 0.8%.

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
