# Peer review of "Pyrolytic Behavior of Major Biomass Components in Waste Biomass"

_polymers, 2019, doi:10.3390/polym11020324_

Round 1

Reviewer 1 Report

The article „Pyrolytic behavior of major biomass components in the waste biomasses“ reports the characterization of pyrolysis oil produced from different sources such as cellulose, hemicellulose, lignin, tannin, pine bark, and pine residues. I consider the idea to compare pyrolytic behavior as well as the composition of pyrolysis oil of the materials mentioned above as excellent. The characterization by HSQC-NMR is suitable and received results have been correctly interpreted.

I would recommend specifying the used method and exact conditions how the kraft pulping liquor has been prepared. The reason is that there are differences if the kraft pulping liquor has been prepared by pulp company  (different stages and dilutions of kraft pulping liquors) or under laboratory conditions (in this case, it would be helpful for readers describe used the methodology in detail not only by citation).

I recommend accepting this paper after the addition of the kraft pulping liquor preparation methodology.

Author Response

We thank reviewer’s advice about this comment and we have added this part to the section of experiment in page 4.

Page4:

2.3. Kraft pulping

The softwood kraft pulping liquor used were prepared using conventional method[31]. Loblolly pine was employed as the wood source for the pulping process. Some relevant preparation conditions are presented in table 1.

Table 1. Conventional pulping conditions[31]

Kraft pulpingConventional
WL Sulfidity,%34.6
EA,%19.7
     Impregnation19.7
Temperature,170
Time,min95

Reviewer 2 Report

Ben et al. studied the Pyrolytic behavior of major biomass components in the waste biomasses.

In their study, they used HSQC-NMR technique to have detailed characterization of the whole portion of pyrolysis oils from biomass including cellulose, hemicellulose, lignin and tannin. The strengths of this technique were relied upon the less spectral overlap and shorter NMR acquisition time. Also, the method provides a complete picture of chemical compounds in pyrolysis bio-oils without any complementary characterization techniques.

The paper is well structured and interesting to read. The experimental methods and results are well written and clearly presented.

It is a nice contribution that meets the standards of Polymers and should be published after minor revision as indicated below:

-          It would be more valuable if the authors include a section on the industrial application of characterized chemical compounds of pyrolysis bio-oils

-          I would suggest citing the recent work of Weckhuysen et al. on pyrolysis bio-oils, DOI: 10.1021/acssuschemeng.6b01329

Author Response

Point 1: It would be more valuable if the authors include a section on the industrial application of characterized chemical compounds of pyrolysis bio-oils.

Response 1: We thank reviewer’s advice about this comment and we have added this section to the paper in page 13.

Page 13:

4. Industrial application of pyrolysis bio-oils

Pyrolysis bio-oils are mixture of about 200 organic compounds. It has been reported that all of the determined compounds classified into nine groups: furans, aldehydes, ketones, phenols, acids, benzenes, alcohols, alkanes, and PAHs[37]. The characterized chemical compounds of bio-oils are used in different industrial fields due to their unique properties[38]. The catechol obtained by pyrolysis of tannin has a very high added value and is widely used in the fields of rubber, electroplating, antiseptic, sterilization, etc. Moreover, furfural, a natural dehydrating product of 5-carbon sugars (e.g. arabinose and xylose) from hemicellulose biomass[39], is regaining attention as a biobased alternative for the production of everything from antacids and fertilizers to plastics and paints[40]. It has been reported that levoglucosan can be used as a specific molecular indicator to trace biomass combustion in sediments[41]. The application of this novel molecular tracer may bring benefits to research areas such as paleoecology, archaeology and environmental science[41]. Furthermore, aromatic hydrocarbons have good rubber compatibility, high temperature resistance, low volatile, and significantly improve the processing properties of rubber. Therefore, it has attracted wide attention in the field of reclaimed rubber and various rubber products.

Point 2: I would suggest citing the recent work of Weckhuysen et al. on pyrolysis bio-oils, DOI: 10.1021/acssuschemeng.6b01329

Response 2: We thank reviewer’s comment and we have addressed this in the revised manuscript by adding it as reference 22.

Page 3:

In a recent study, quantitative 13C NMR combined with comprehensive two-dimensional gas chromatography (GC × GC) was used to characterize fast pyrolysis bio-oils, which provided new information on the chemical composition of bio-oils for further upgrading[22].
